# Supervised Segmentation of NO$_2$ Plumes from Individual Ships Using TROPOMI Satellite Data

**Solomiia Kurchaba** [1,*] **, Jasper van Vliet** [2]**, Fons J. Verbeek** [1]**, Jacqueline J. Meulman** [3] **and Cor J. Veenman** [1,4]

1    Leiden Institute of Advanced Computer Science (LIACS), Leiden University,
     2333 CA Leiden, The Netherlands
2    Human Environment and Transport Inspectorate (ILT), 3531 AH Utrecht, The Netherlands
3    Mathematical Institute, Leiden University, 2333 CA Leiden, The Netherlands
4    Data Science Department, TNO, 2595 DA The Hague, The Netherlands
*    Correspondence: s.kurchaba@liacs.leidenuniv.nl

**Abstract:** The shipping industry is one of the strongest anthropogenic emitters of NO$_x$—a substance harmful both to human health and the environment. The rapid growth of the industry causes societal pressure on controlling the emission levels produced by ships. All the methods currently used for ship emission monitoring are costly and require proximity to a ship, which makes global and continuous emission monitoring impossible. A promising approach is the application of remote sensing. Studies showed that some of the NO$_2$ plumes from individual ships can visually be distinguished using the TROPOspheric Monitoring Instrument on board the Copernicus Sentinel 5 Precursor (TROPOMI/S5P). To deploy a remote-sensing-based global emission monitoring system, an automated procedure for the estimation of NO$_2$ emissions from individual ships is needed. The extremely low signal-to-noise ratio of the available data, as well as the absence of the ground truth makes the task very challenging. Here, we present a methodology for the automated segmentation of NO$_2$ plumes produced by seagoing ships using supervised machine learning on TROPOMI/S5P data. We show that the proposed approach leads to more than a 20% increase in the average precision score in comparison to the methods used in previous studies and results in a high correlation of 0.834 with the theoretically derived ship emission proxy. This work is a crucial step towards the development of an automated procedure for global ship emission monitoring using remote sensing data.

**Keywords:** TROPOMI/S5P satellite; NO$_2$; maritime shipping; supervised learning; remote sensing application; ship plume segmentation

## 1. Introduction

The international shipping sector is one of the strongest sources of anthropogenic emission of NO$_x$—a substance that has a negative impact both on ecology and human health. The contribution of the shipping industry is estimated to vary from 15% to 35% worldwide [1,2], which leads to approximately 60,000 premature deaths annually [3]. While over the last 20 years, the pollution produced by power plants, the industry sector, and cars has been constantly decreasing, the impact of maritime transport continues to grow [4]. This causes a big societal pressure, resulting in regulations [5] that put restrictions on emission levels that can be produced by individual ships.

All methods currently used for ship emission monitoring such as in situ [6,7], on-board [8], and airborne-platform-based [9] have several disadvantages: they all require close proximity to a ship; they are costly; they do not allow for monitoring on a global scale. A potential solution to the problem is the application of remote sensing instruments [10]. Studies [11,12] show that NO$_2$ traces of some individual ship plumes can be visually distinguished on images from the TROPOspheric Monitoring Instrument on board the Copernicus Sentinel 5 Precursor (TROPOMI/S5P) satellite [13], which was launched in 2018.

An important next step required for the development of an automated global-scale monitoring system is an automated interpretable method for the evaluation of $NO_x$ emission produced by individual ships. Among the main challenges for this development are a low temporal sampling rate and spatial resolution, resulting in an extremely low signal-to-noise ratio. In addition, there is a high risk of interference of the ship plume with other $NO_x$ sources and a high frequency of occurrence of plume-like objects that cannot be associated with any ship. Finally, the ground truth for this task is not available. In this paper, we present a methodology that allows addressing the above-mentioned challenges. Using machine learning and taking advantage of the spatial characteristics of a ship plume, the developed method allows automatic segmentation of $NO_2$ plumes from the background, simultaneously assigning the detected signal to a ship-emitter, circumventing the listed limitations.

We used $NO_2$ retrievals from the TROPOMI/S5P as this is the only available remote sensing instrument that performs $NO_2$ measurements with a resolution high enough (The ground pixel resolution of the TROPOMI/S5P instrument equals $3.5 \times 5.5$ km$^2$ at nadir) to distinguish plumes from individual ships. To increase the number of potentially distinguishable plumes, we enhanced the contrast between the ship plumes and the background. The used enhancement technique allows for a differentiation between the ship plumes and random co-occurring concentration peaks in the ships' neighborhood. The application of the image enhancement technique also allows for an improvement of the low signal-to-noise ratio. Then, for each analyzed ship, we performed an automatic generation of a Region Of Interest (ROI) that we call a *ship sector*. The purpose of the *ship sector* is to focus the area of analysis on the region where the ship plume is expected to be located. Subsequently, we normalized the *ship sector* and divided it into sub-regions. This way, we distinguish the plume of interest from all the other $NO_2$ plumes or land-origin outflows that potentially might be located within the *ship sector*. Based on the *ship sector* division, we created a set of spatial features that characterize the location of the $NO_2$ plume within the *ship sector*. Due to the absence of other sources of ground truth, each pixel of the *ship sectors* we manually labeled as a "plume" or "not a plume". Trained on the manually labeled data, a machine learning model enabled us to automatically segment plumes in unseen images. We studied five robust machine learning models of increasing complexity and compared their performance with the threshold-based methods used in previous studies. To validate the developed pipeline, we compared the estimated, based on the segmentation results, amount of $NO_2$ to the theoretically derived ship emission proxy [11].

The rest of this paper is organized as follows: In Section 2, we start with an overview of the related literature. We then provide a description of the data sources used in this study and introduce the developed methodology (Section 3). In Section 4, the reader can find the results of the study, which are followed by the conclusions in Section 5 and discussion in Section 6.

## 2. Related Work

For more than a decade, scientists have been trying to use the available satellite data in order to quantify the $NO_2$ emission from the shipping industry as a whole. For instance, using the measurements from the Global Ozone Monitoring Experiment (GOME) [14] instrument on board the second European Remote Sensing satellite (ERS-2), the authors estimated the $NO_2$ emission level above the shipping lane between Sri Lanka and Indonesia [15]. With the images from the SCanning Imaging Absorption spectroMeter for Atmospheric CartograpHY (SCIAMACHY) [16] on board the ENVIronmental SATellite (Envisat) mission, traces from the shipping industry over the Red Sea were quantified [17]. Finally, data from the Ozone Monitoring Instrument (OMI) [18] aboard the NASA Aura spacecraft was used to visualize a ship's $NO_x$ emission inventory for the Baltic Sea [19]. However, due to the significantly lower resolution capabilities of the above-mentioned predecessors of TROPOMI/S5P (GOME: $40 \times 320$ km$^2$, SCIAMACHY: $30 \times 60$ km$^2$, OMI:

$13 \times 25$ km$^2$, TROPOMI: $3.5 \times 5.5$ km$^2$), these studies were based on multi-month data averaging, which does not give the possibility to quantify the emission from individual ships.

There are many studies demonstrating the capability of TROPOMI NO$_2$ measurements to pinpoint the emission from urban (e.g., [20–22]) or industrial sources, such as the mining industry [23] or a gas pipeline [24], as well as showing the possibility of the usage of TROPOMI data to quantify the positive effects of the COVID-19 lockdowns (e.g., [25]). Nonetheless, all studied emission sources are stationary (therefore, can be observed over an extended period of time) and emit much higher quantities of NO$_x$ than an individual ship. Thus, all the above-mentioned problems are arguably less complex than the one discussed in this paper.

In [11], it was reported that the NO$_2$ plumes produced by individual ships can be visually distinguished in TROPOMI data. However, since the NO$_2$ traces of most of the ships in the area are not sufficiently stronger than the background concentrations, only plumes of the largest ships were addressed in that study. In addition, the presented approach requires multiple manual steps, which makes it impossible to apply the method on a global scale. In [12], we introduced the first attempt of a fully automated pipeline for the estimation of NO$_2$ emission from individual ships. In the study, we showed that pre-processing of the TROPOMI signal allows a visual distinction of a greater amount of ship plumes. The plume–background separation itself, however, was based on a locally optimized threshold, established individually for each of the analyzed ships. Due to the fact that the threshold was established on the basis of the only variable (NO$_2$ concentration), it was not flexible enough for a good quality of separation of the ship plume from the background. Finally, the one-feature-based method of thresholding does not allow for differentiation of the plume produced by the analyzed ship from all the other NO$_2$ plumes that might be located in the ship's proximity.

Machine learning has proven to be an efficient technique for solving problems in geosciences and remote sensing [26,27]. Recently emerging studies show the efficiency of applying machine learning models to the TROPOMI data as well. For instance, in [22], the authors built a deep convolution neural network to classify images into those that contain an NO$_2$ plume from those that do not. The developed model, however, does not differentiate the sources of the detected plumes. Moreover, the study does not provide the attempts of the segmentation of the detected plumes from the background; thus, there is no possibility to quantify the amount of NO$_2$ that was emitted by a given source. Several studies reported a successful application of various multivariate machine learning models for the estimation of surface-level emissions. For example, in [28], the authors created a multivariate machine learning model for the estimation of NO$_2$ emission over Germany [28], while in [29], the O$_3$ concentrations were estimated for California. The areas of analysis of the above-mentioned studies, however, are restricted to over-land territories. In [30] was reported the first attempt to extend near-surface concentration predictions to an ocean region. The authors acknowledged that the performance of emission estimations over the ocean is significantly more challenging than the equivalent task for the over-land areas, mainly due to the absence of in situ measurements' possibilities. In addition, the sources of the detected emission levels have not been studied in the above-mentioned paper.

In this study, we present a pipeline that, for the first time, allows the application of multivariate machine learning models for the estimation of NO$_2$ emission from seagoing ships. The developed method uses TROPOMI satellite data, AIS data on ship positions, as well as ECMWF wind data for the segmentation of NO$_2$ plumes from individual ships, allowing differentiation between the plume produced by the ship of interest from all the other NO$_2$ plumes in the ship neighborhood. In the following section, the data sources used and the developed methodology are presented in detail.

## 3. Materials and Methods

### *3.1. Data Sources*

In this paper, we performed the integration of several data sources: TROPOMI satellite measurements, wind data, information about the ships' positions, and information about the properties of analyzed ships. In this section, the reader can find a general overview of the data sources that will be utilized throughout the study. We also provide here the criteria applied for data selection, which set the scope of this study.

#### 3.1.1. TROPOMI Data

TROPOspheric Monitoring Instrument (TROPOMI) [13] is a UV–Visible–Near-Infrared–Short-Wave Infrared (UV–Vis–NIR–SWIR) spectrometer on board the Copernicus Sentinel 5 Precursor (S5P) satellite. The satellite was launched in October 2017 and entered its operational phase in May 2018. It is a Sun-synchronous satellite with a local equatorial overpass time at 13:30. In 24 h, the satellite performs approximately 14 orbits and, with these, covers the full globe.

The TROPOMI spectrometer measures spectra of several trace gases including nitrogen dioxide ($NO_2$). Since $NO_2$ gas is the most notable product of photochemical reactions of $NO_x$ emitted by ships, it can be utilized for the ships' emission monitoring. The maximal ground pixel resolution of the TROPOMI instrument is equal to $3.5 \times 5.5$ km$^2$ at nadir. Due to the projection of the satellite images, the real size of the pixel will vary, depending on the true distance between the satellite and the part of the surface of the Earth being imaged. To generate images of regular size, we regridded (For the data regridding, the HARP v.1.13 Python package was used) the original TROPOMI data into a regular-sized grid of size $0.045° \times 0.045°$, which for the pixel in the middle of the analyzed area translates to approximately $4.2 \times 5$ km$^2$.

The following filtering criteria were applied to the TROPOMI data: *qa_value* > 0.5, *cloud fraction* < 0.5. Previous studies [11,12] showed that such a selection of filtering values can yield considerably good results for a given task. The detailed description of the variables can be found in [31].

In this study, 68 days from the period between 1 April 2019 and 31 December 2019 of TROPOMI measurements were analyzed. The analyzed days were mostly sunny—the distribution of the variable *cloud fraction* for the scope of this study is provided in Figure 1. The studied data product was the tropospheric vertical column of nitrogen dioxide [32], data version: 1.3.0. For the analysis, an area in the Mediterranean Sea, restricted by the Northern coasts of Libya and Egypt from the south and the south coast of Crete from the north (lon: [19.5°; 29.5°], lat: [31.5°; 34.2°]) was chosen. This particular region was selected because of the presence of a busy shipping lane connecting Europe and Asia, the high frequency of occurrence of sunny days, and relatively low levels of $NO_2$ background concentrations, which are favorable conditions for the analysis. An outline of the area studied is presented in Figure 2.

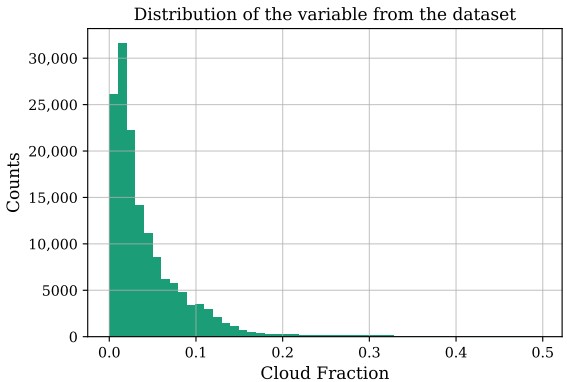

**Figure 1.** Distribution of the variable *cloud fraction* for the dataset used in this study.

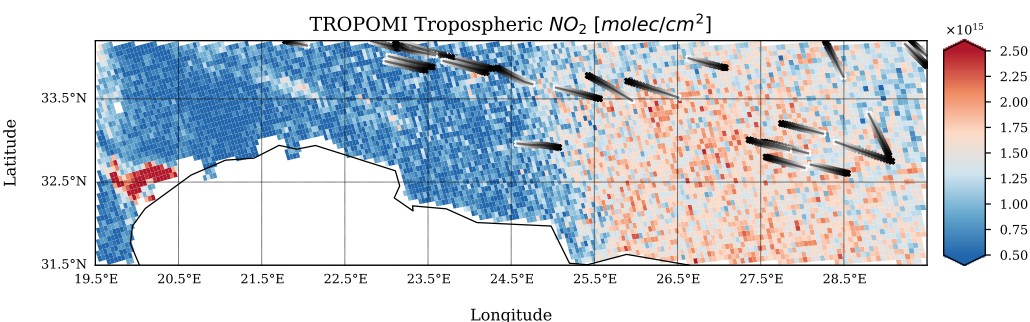

**Figure 2.** The NO$_2$ tropospheric column. Visualized day: 14 June 2019. Studied area: the Mediterranean Sea, restricted by the northern coasts of Libya and Egypt from the south and the south coast of Crete from the north. Black lines indicate ships' tracks based on information from AIS data. The red area on the right-hand side of the figure corresponds to the land outflow from the variety of land base sources of NO$_2$. For the convenience of visualization, the presented data were not regridded—the native local size of the TROPOMI pixels are presented in the figure.

### 3.1.2. Wind Data

In this study, we used wind data from the European Center for Medium range Weather Forecasts (ECMWF). The wind fields (wind speed and wind direction) were taken from the ECMWF operational model analyses at a spatial resolution of 0.25° (For the analyzed area, a spatial resolution of 0.25° × 0.25° translates to ≈23.4 × 27.6 km$^2$), a temporal resolution of 6 h, and an altitude of 10 m. It was shown in [11] that wind data at a 10 m altitude are an optimal choice for the task of ship–plume matching. Starting from the product version upgrade from 1.2.2 to 1.3.0, which took place on 27 March 2019, the ECMWF 10 m wind data for coinciding time are available as a support product in the TROPOMI/S5P data file.

### 3.1.3. Ship-Related Data

Since 2002, all commercial seagoing vessels are obliged to carry on board an AIS transponder [33], which transmits information about position, speed, heading (direction), and a unique identifier (MMSI) and the type of each ship. Thus, another source of data used in this study is data from Automatic Identification System (AIS) transponders. At the moment, there is no open-access AIS data available for the region and time period, as described in Section 3.1.1, as well as the quality required for this study. The data, however, can be accessed through several commercial providers. For the scope of this study, the AIS data, as well as information about the dimensions of the ships were provided by the Netherlands Human Environment and Transport Inspectorate (ILT). This is the Dutch national designated authority for shipping inspections, has access to commercial databases for the AIS dataset used in this study, and is participating in this research.

With the aim of the reduction of the number of images where the ship plume cannot be visually detected, in our study, we only focused on ships with a speed that exceeds 14 kt. If two ships move in immediate proximity to each other, only the ship with the highest speed was taken into consideration. From the analysis were also excluded ships that are not involved in global trade, such as yachts, leisure vessels, or research vessels. In Figure 3, the information about the dates used for this study, as well as the number of ships per day studied is depicted. The differences in the number of ships per studied day can be caused by bad weather conditions on the measurement day.

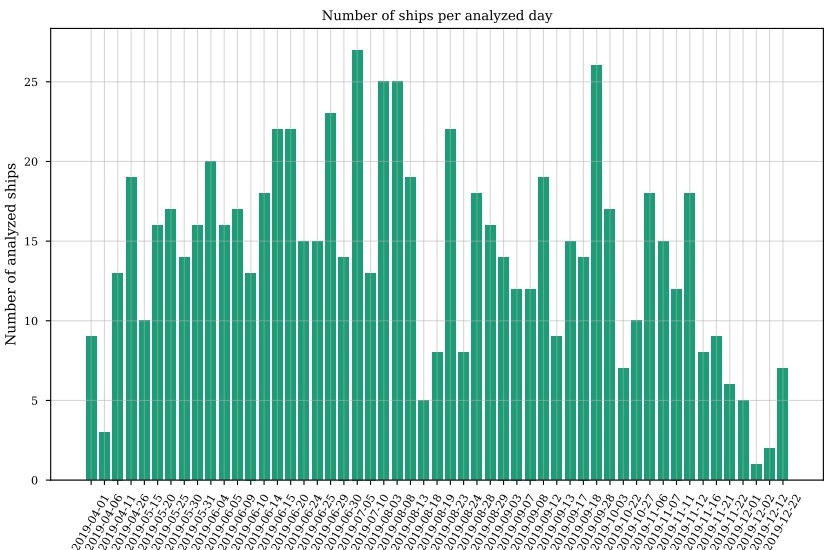

**Figure 3.** A list of days used for the dataset creation and the number of ships per day studied.

*3.2. Method*

In this subsection, we present the developed methodology. Taking advantage of the characteristics of the analyzed ship, as well as wind conditions in the studied region, our approach allows the segmentation of the $NO_2$ plume produced by the particular ship of interest, distinguishing it from all the other concentration peaks in the surrounding area. The results produced by the proposed approach are easily interpretable and, thus, can be used as a reliable source of information by ship inspectors.

The method consists of the following steps: an AIS data-based interpolation of the ship tracks at the moment and just before the satellite overpass (Section 3.2.1), definition (Section 3.2.2), and enhancement (Section 3.2.3) of a ship plume image, the definition of a ship sector (Section 3.2.4) that allows the further restriction of the analyzed area, normalization of the defined ship sector, and splitting of the normalized sector into sub-regions (Section 3.2.5), which, finally, gives the possibility to retrieve the set of necessary features. These steps are described below.

3.2.1. Ship Tracks

The first step is to estimate the tracks of the studied ships. Taking into account a lifetime of $NO_x$ equal to a few hours, we studied the track of the ship starting from two hours before the moment of the satellite overpass. We distinguished two types of ships' tracks: the *ship track*, obtained based on resampling and interpolation of the AIS data, and the *wind-shifted ship track*. For the calculation of the *wind-shifted ship track*, we assumed that the plume emitted by a ship has moved in accordance to the wind direction by a distance $d = v \times |\Delta t|$, where $v$ is the speed at a given location at 10 m above sea level from ECMWF for 12:00 UTC, and $|\Delta t|$ is the time difference between the time of the satellite overpass and the time of a given AIS ship position. Both wind speed and wind direction are assumed to be constant for the whole time during which we study the plume. Such an assumption may create uncertainties in the expected position of the plume of the ship. However, with the methodology presented further in this section, such uncertainties will not affect the correctness of the ship-plume allocation.

Summing up, the *ship track* provides us the information on the position of the ship from where the studied ship plume was emitted. The *wind-shifted ship track* indicates the expected position of the center line of the $NO_2$ plume after displacement driven by local wind conditions. Figure 4a,b give examples of the *ship track* and its corresponding *wind-shifted ship track*, respectively.

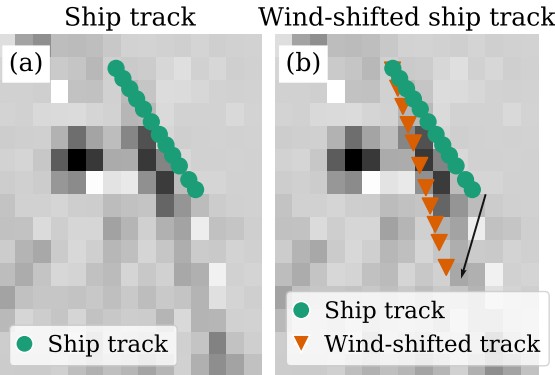

**Figure 4.** (**a**) *Ship track*—estimated, based on AIS data records. The ship track is shown for the time period starting from 2 h before the satellite overpass until the moment of the satellite overpass. (**b**) *Wind-shifted ship track*—a ship track shifted in accordance with the speed and direction of the wind. The *wind-shifted ship track* indicates the expected position of the ship plume. A black arrow indicates the wind direction. For both presented images, the size of the pixel is equal to $4.2 \times 5$ km$^2$.

### 3.2.2. Ship Plume Image

Utilizing the knowledge of a ship's position summarized in its *ship track* and *wind-shifted ship track*, we are able to focus our attention on the area that lies within immediate proximity to the analyzed ship. For this, the concept of a *ship plume image* (see Figure 5a) is introduced. The area of the *ship plume image* is determined based on the *wind-shifted ship track* as follows: the average coordinate of the studied *wind-shifted ship track* defines the center ($longitude_{centr}, latitude_{centr}$) of the *ship plume image*; the borders of the image are defined as $longitude_{centr}, latitude_{centr} \pm 0.4°$ (For the area in the Mediterranean Sea, in horizontal direction $0.4° \approx 37.4$ km, in vertical direction $0.4° \approx 44.2$ km). This particular size of a *ship plume image* was determined in order to allow for optimal plume coverage for the most typical range of ship speeds (14 kt–20 kt) (kt—knot, a unit of speed equal to a nautical mile per hour; 14 kt $\approx$ 26 km/h; 20 kt $\approx$ 37 km/h). Given the size of the pixel grid, such an offset results in an image of a maximal dimension of $18 \times 18$ pixels.

### 3.2.3. Pre-Processing of a Ship Plume Image

To improve the quality of the satellite signal, in the data pre-processing step, on each of the analyzed *ship plume images*, we applied the local Moran's *I* spatial auto-correlation statistic [34]. In [12], we showed that the application of this technique substantially improves separability between the ship plume and the background.

The local Moran's *I* spatial auto-correlation statistic allows the enhancement of the intensity of high-value pixels located in a cluster while suppressing isolated concentration peaks randomly occurring in the background. We characterized a ship plume as a cluster of pixels adjacent to each other with a concentration higher than the background average. This way, calculating the spatial auto-correlation of a *ship plume image*, we combined image denoising with the enhancement of the relevant part of the image.

An example of a result of an *enhanced ship plume image* is provided in Figure 5b.

Formally, Moran's *I* spatial auto-correlation statistic is defined as follows: for each pixel *i* of a *ship plume image*, the local Moran's *I* is calculated as:

$$I_i = \frac{(x_i - \mu)}{\sigma^2} \sum_{j=1, j \neq i}^{N} w_{ij}(x_j - \mu), \tag{1}$$

where $x_i$ is the value of the respective pixel, *N* is the number of analyzed pixels of a *ship plume image* (in our case, $18 \times 18$), $\mu$ is the mean value of all *N* pixels, $\sigma^2$ their variance, and $w_{ij}$ is the value of an element in a binary spatial contiguity weight matrix *W* at location *j* with regard to the analyzed pixel *i*. The value of an element of the binary spatial contiguity matrix $w_{ij}$ is 1 for pixels that are considered to be the neighbors of the analyzed pixel *i*, and

0 otherwise. For the study, the queen spatial contiguity [35], which is the $3 \times 3$ 8-connected neighborhood of the analyzed (central) pixel, was applied. The value $I_i$ becomes the value of the corresponding pixel of the resulting enhanced image.

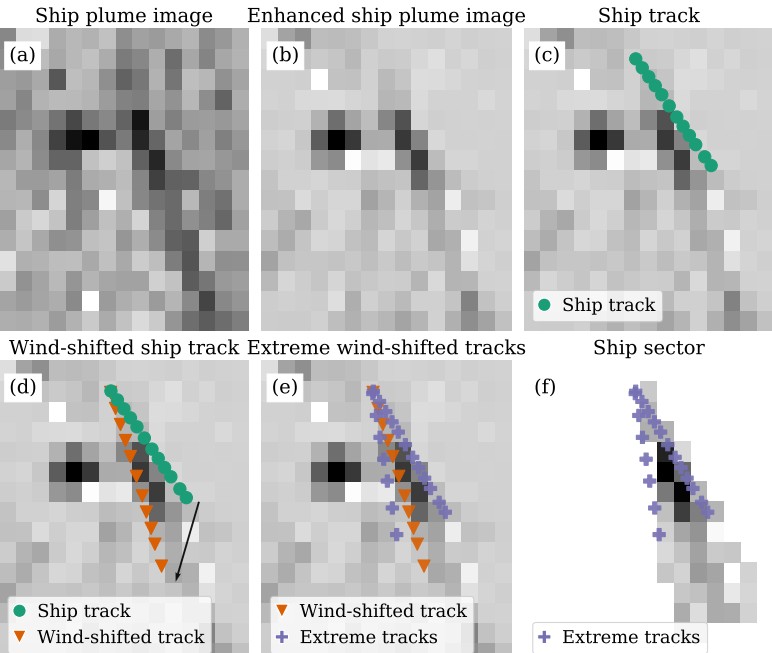

**Figure 5.** Ship sector definition pipeline. (**a**) *Ship plume image*—the TROPOMI NO$_2$ signal for the area around the analyzed ship. Two ship plumes can be distinguished, but only one is of interest. (**b**) The NO$_2$ signal enhanced by Moran's *I* spatial auto-correlation statistic. (**c**) *Ship track*—estimated, based on AIS data records. The ship track is shown for the time period starting from 2 h before the satellite overpass until the moment of the satellite overpass. (**d**) *Wind-shifted ship track*—a ship track shifted in accordance with the speed and direction of the wind. The *wind-shifted ship track* indicates the expected position of the ship plume. A black arrow indicates the wind direction. (**e**) *Extreme wind-shifted ship tracks*—calculated, based on wind information with assumed uncertainties; define the borders of the *ship sector*. (**f**) A resulting *ship sector*—an ROI of an analyzed ship. For all presented images, the size of the pixel is equal to $4.2 \times 5 \, \text{km}^2$.

From Equation (1), we can imply the weak side of using Moran's *I* statistic for ship plume enhancement: apart from the clusters of high values, the given method enhances clusters of low-value pixels at the same time. The methodology proposed in this study, however, is designed in a way so that this negative impact is minimized.

### 3.2.4. Ship Sector

A plume produced by a ship at a given moment will be displaced, over time, in the direction of the wind in the analyzed area. Having the wind information available, we would like to restrict the analysis to the part of the *ship plume image*, where the probability to find the plume of the ship is the highest. We performed the area restriction by defining an ROI of an analyzed ship, which we call a *ship sector*. The area of the *ship sector* is determined on the basis of information about the ship's trajectory and the speed/direction of the local wind.

As a starting point of the *ship sector's* definition, we used the *wind-shifted ship track*, calculated as described in Section 3.2.1. We then calculated the *extreme wind-shifted tracks* by adding the margin of wind-related uncertainty to either side of the *wind-shifted ship track*. The *extreme wind-shifted tracks* delineate the borders of the *ship sector*, showing the extreme possible positions of the plume, taking the wind measurement uncertainty into account. The wind's uncertainty was assumed due to the limited spatial and temporal resolution of wind data [11], based on reported in several studies' [36,37] measurement bias, as well

as assumptions of constant wind mentioned in Section 3.2.1. Illustrations of the *extreme wind-shifted tracks* and the resulting *ship sector* are shown in Figure 5e and 5f, respectively. As a result of the delineation of the *ship sector* area, the plume should always lie within the *ship sector* boundaries. Only pixels lying within the *ship sector* were taken into consideration in further analysis. Parameters related to the *ship sector* can be found in Table 1.

**Table 1.** Parameters applied for ship sector definition.

| Parameter | Value |
|---|---|
| Trace track duration | 2 h |
| Wind speed uncertainty | 5 m/s |
| Wind direction uncertainty | 40° |

### 3.2.5. Feature Set Construction

In order to obtain a multivariate description of the *ship sector* pixels, we encoded the spatial information into a set of generic features. First, we performed a *ship sector* normalization to make spatial information in the sector comparable between the different sectors. We defined a *normalized sector* by the standardization of the orientation and the scale of the original *ship sector*. In this way, the position of the plume within the *ship sector* becomes invariant to the heading (direction) and speed of the ship, as well as to the direction and speed of the wind.

We standardized the orientation of a *ship sector* by rotating to 320° so that the angle of the polar coordinate of the corresponding *wind-shifted ship track* is the same for all ships (see Figure 6). The particular value of the sector rotation angle was chosen for the convenience of visualization and has no influence on further modeling. Assuming $S$ is a set of ship sectors in the dataset, formally, the rotation coordinates of a *ship sector* are defined in the following way:

$$\forall s \in S, \quad \forall i \in s: \quad lon\_rotated_{s,i} = r_{s,i} \cdot cos(\alpha_{s,i} + \Theta_s), \quad lat\_rotated_{s,i} = r_{s,i} \cdot sin(\alpha_{s,i} + \Theta_s), \quad (2)$$

where $lon\_rotated_i$ and $lat\_rotated_i$ are the polar coordinates of the pixel $i$ within the rotated *ship sector*, $r_{s,i}$ is the radial distance of the pixel $i$ from the origin of the *ship sector* $s$ (in our case, the sector origin corresponds to the position of the ship at the moment of satellite overpass), $\alpha_{s,i}$ is a counterclockwise rotation angle of the pixel $i$ from the axis $x$ (*longitude*) of the *ship sector* $s$, $\Theta_s = \beta - \alpha_s$ is a counterclockwise rotation angle that will be applied for the orientation change of each pixel $i$ of the *ship sectors*, $\alpha_s$ is a rotation angle of a *ship sector* $s$ that corresponds to the counterclockwise rotation angle of the pixel $i_{s,max}$ with the radial distance from the origin $r_{s,max} = max(r_s)$, and $\beta = 320°$ is a new rotation angle of each *ship sector* $s$ after the rotation.

We standardize the *ship sector's* scale so that the horizontal and vertical coordinates of the rotated *ship sector* are rescaled into the range [0, 1] by applying a min–max scaler on the horizontal and vertical coordinates of the pixel:

$$lon\_norm = \frac{lon\_rotated - min(lon\_rotated)}{max(lon\_rotated) - min(lon\_rotated)},$$

$$lat\_norm = \frac{lat\_rotated - min(lat\_rotated)}{max(lat\_rotated) - min(lat\_rotated)} \quad (3)$$

The second step of the feature construction procedure is the division of the *normalized sector* into a set of sub-regions that enable encoding spatial information of the pixels within the *normalized sector*. First, we define *levels* of the *normalized sector* by splitting it into six sub-regions on the basis of the radial distance of the pixel from the origin of the sector. Then, we define *sub-sectors* by splitting the *normalized sector* into four sub-regions on the basis of the pixel's rotation angle. As a result, the position of each pixel within the *normalized sector*

image can be characterized in terms of two values: a *level* and a *sub-sector*. An illustration of the *normalized sector* divided into a set of *levels* and *sub-sectors* is presented in Figure 7.

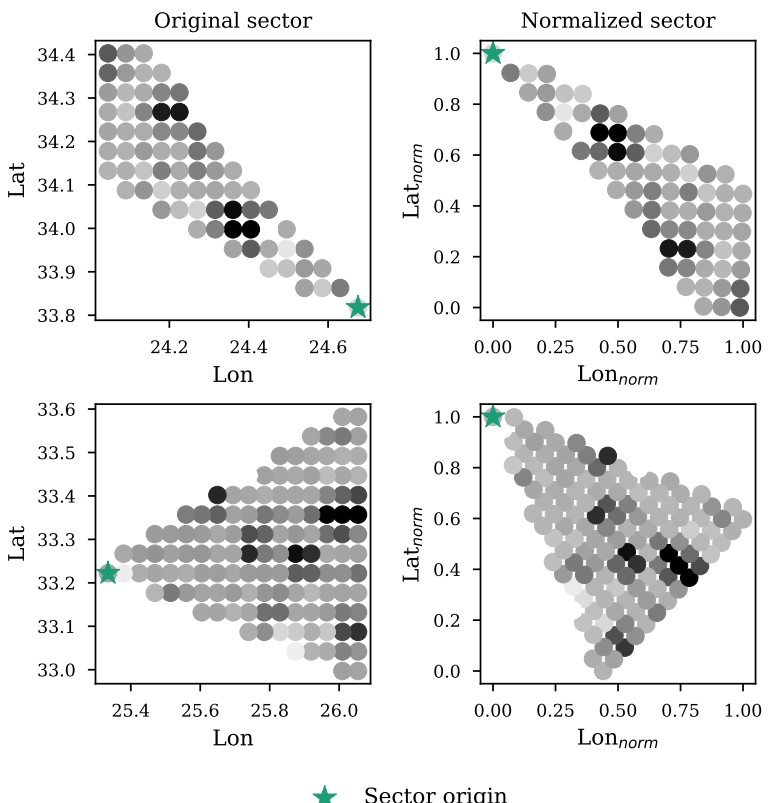

**Figure 6.** Sector normalization. We rotate the *ship sectors* so that all resulting sectors have the same orientation equal to 320°, independently of the original direction of the ship's heading. We then rescaled the image so that the range of both coordinates is between 0 and 1. The gray area in each figure indicates a *ship sector*. The *ship sector* origin indicator shows the position of the ship at the moment of the satellite overpass. Two examples of original and rotated sectors are shown: one in the top row and one in the bottom row.

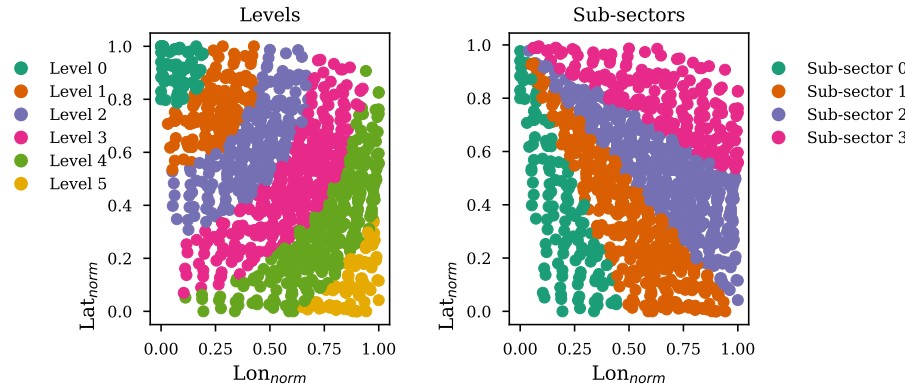

**Figure 7.** Levels and sub-sectors. We perform a feature construction by dividing the *normalized sector* into sub-regions: *levels* and *sub-sectors*. For the convenience of visualization, data points from one day of analysis were used for the preparation of the figure.

### 3.3. Experiment Design

Here, we describe the experimental setup used in this study: first, we describe the dataset used for the training of the multivariate models, then we explain the models used for the benchmarking and provide a list of used multivariate classifiers. In addition, in this

section, the reader can find the description of the methods used for the hyperparameters' optimization and measures utilized for the performance evaluation.

### 3.3.1. Dataset Composition

Following the steps provided in the previous subsections, we created 754 images and cropped them to an area of the *ship sector*. The *ship sector* images were enhanced by Moran's *I* operator and manually labeled so that they can be used for training machine learning models. Not all *ship sector* images contained a visually identifiable NO$_2$ plume. Moreover, due to the dispersion and chemical transformation of a ship plume, some parts of the plume will always be under the detection limit of the satellite and, therefore, indistinguishable. Thus, labeling errors are possible. To minimize the chance of mistakes, the labeler was supported with several representations of the area of interest: the original not enhanced NO$_2$ tropospheric vertical columns for the area of a *ship plume image*, the enhanced with the Moran's *I* area of a *ship plume image*, and NO$_2$ tropospheric vertical columns for the full studied area in Mediterranean Sea with the positions of the neighboring ships. The descriptive statistics of the resulting dataset are provided in Figure 8. In Table 2, the information on the data distribution within the two classes of the dataset is shown. All mentioned numbers correspond to the full dataset before the training/test set division. The dataset used in this study, along with the code used for the experiments can be found under the following link: https://github.com/SolaK24/ShipPlumeSegmentation_Supervised 28 September 2022.

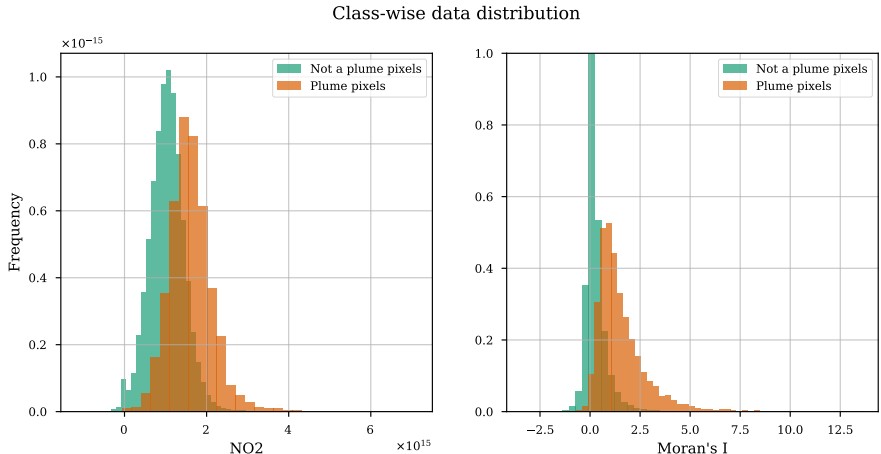

**Figure 8.** Classwise distribution of the two main features of the dataset: NO$_2$ and Moran's *I*.

**Table 2.** The number of measurement points per class in the dataset.

|                    | No Plume | Plume |
| ------------------ | :------: | :---: |
| Number of pixels   | 68,646   | 6980  |
| Number of images   | 208      | 535   |

### 3.3.2. Multivariate Models

To exploit the potential of multivariate modeling, we used several classifiers of increasing complexity: Logistic Regression, Support Vector Machine with a linear kernel [38], Support Vector Machine with a radial basis kernel [39], Random Forest [40] (All above-mentioned models were implemented in the Scikit-learn v. 0.24.2 package [41]), and Extreme Gradient Boosting (XGBoost) [42] (Implemented in the xgboost Python package v. 1.3.3). All the models are multivariate and, thus, are able to benefit from the set of prepared features, namely the set of spatial features developed with the method described in Section 3.2.5, along with ship and wind-related features. All models selected for the experiment are highly robust. Therefore, the potential mistakes in human labeling, if present in reasonable amounts, should still allow for the models' proper training.

The first feature of the model is enhanced by *Moran's I* values of the pixels that were translated into a one-dimensional feature vector. As mentioned in Section 3.2.3, the application of Moran's *I* may result in the creation of additional high-value pixels resulting from the enhancement of clusters of low-value pixels. To mitigate the negative impact of this side effect, apart from Moran's *I*, the feature set was composed of the corresponding value of $NO_2$. This way, a supervised learning model will be able to differentiate between high- and low-value enhanced $NO_2$ clusters. Other features used by the model are *Wind Speed*, *Wind Direction* (encoded into its sine and cosine components, in order to enable a continuous feature space for various wind directions), *Ship Speed*, and *Ship Length*. Finally, the position of an analyzed pixel within the *normalized sector* in terms of *levels* and *sub-sectors* was translated into the feature vectors using one-hot encoding. The resulting feature set was composed of 17 features in total. For the full feature list, see Figure in Section 4.1. The used binary label indicates whether the given pixel is a part of the ship plume or not.

### 3.3.3. Benchmarks

To quantify the performance improvement gained by the usage of multivariate supervised models, we performed ship plume segmentation by applying a thresholding method on a single selected feature. First, we applied a thresholding method on the tropospheric vertical column of the $NO_2$ TROPOMI product regridded in accordance with the description in Section 3.1.1. No image enhancement technique was applied. This simplest way of plume-background separation was used, among others, in [17] for the quantification of $NO_2$ emission from the international shipping sector. In [11], the separation of pixels related to $NO_2$ plumes from individual ships was also performed based on solely TROPOMI $NO_2$ data. In this paper, we refer to this benchmarking method *NO₂ threshold*. Visualization of the input data for this thresholding technique can be found in Figure 9a.

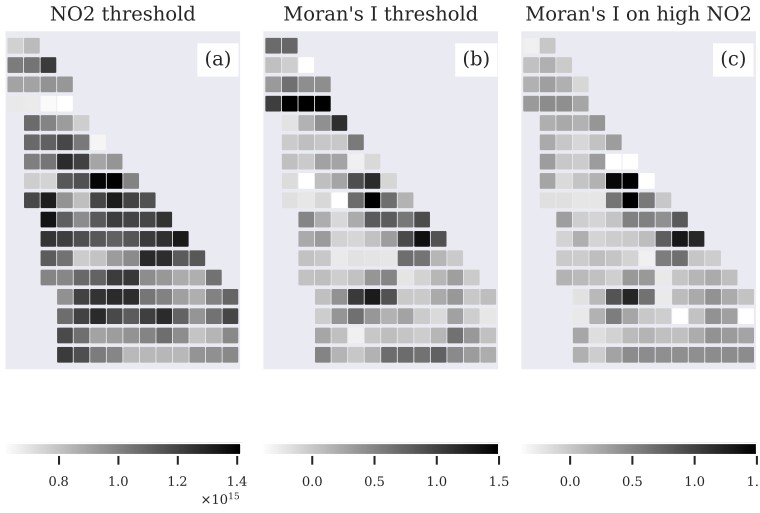

**Figure 9.** Input data example for univariate threshold-based benchmarks. (**a**) Input data for a benchmark method *NO₂ threshold*. (**b**) Input data for a benchmark method *Moran's I threshold*. At the top of the *ship sector*, the reader can find an example when a cluster of low-value $NO_2$ was mistakenly enhanced by Moran's *I*. (**c**) Input data for a benchmark method, *Moran's I on high NO₂*. For all presented images, the size of the pixel is equal to $4.2 \times 5$ km$^2$.

As a second benchmarking method, following the suggestion made in [12], we performed a ship plume segmentation based on images enhanced with Moran's *I* statistic. The satellite image enhancement allows effective separation of a greater amount of $NO_2$ plumes. However, as mentioned in Section 3.2.3, the application of Moran's *I* statistic may result in the enhancement of low-value clusters that are not a part of a plume. Visualization of the input data for this benchmarking technique is presented in Figure 9b. In the rest of the article, we call this method *Moran's I threshold*.

To overcome the problem of the enhancement of low-value clusters by Moran's *I*, we propose to assign zero value to all pixels of the image with intensity lower than the median of the given *ship sector* picture and afterward apply the Moran's *I* enhancement. This is the third benchmarking method used in this study. We call it *Moran's I on high NO$_2$*. Visualization of the results of the application of Moran's *I* only on high NO$_2$ values can be found in Figure 9c. As presented in Figure 9, for all three benchmarking methods, only pixels that lie within the *ship sector* area were taken into account for segmentation.

### 3.3.4. Segmentation Validation Metrics

For the assessment of classification quality, we used a precision–recall curve, an Average Precision score (AP), which is defined as the Area Under the Precision–Recall Curve (PR-AUC), a Receiver Operating Curve (ROC), and, finally, the Area Under the Receiver Operating Curve (ROC-AUC). All evaluation methods were implemented in the Scikit-learn v. 0.24.2 package [41].

Precision and recall are respectively defined as follows:

$$Precision = \frac{TP}{TP + FP} \tag{4}$$

$$Recall = \frac{TP}{TP + FN}, \tag{5}$$

where *TP* stands for true positive and corresponds to the pixels that were labeled as a "plume" and were correctly identified by the classifier. *FP*—false positive; it corresponds to the pixels that were not labeled as a "plume", but were identified as a "plume" by the classifier. *FN* stands for false negative and corresponds to the pixels that were not classified as a "plume" by the classifier, but were labeled as such by the labeler. The ROC curve visualizes True Positive (*TP*) scores as a function of False Positive (*FP*) scores.

### 3.3.5. Cross-Validation and Parameters' Optimization

For the model fine-tuning and model performance evaluation, nested cross-validation [43,44] was used. In the inner loop, we performed a randomized grid-search [45] with 5-fold cross-validation to optimize the hyperparameters of the used models. The AP score was used as a target function for optimization. The performance of the best model identified during the inner loop of cross-validation was evaluated on the "hold-out" test set, generated during the outer loop of cross-validation. The above-mentioned procedure was repeated five times, generating five independent training and test sets. An illustration of the applied cross-validation scheme can be found in Figure 10. The search space of the hyperparameters for each of the analyzed multivariate models is provided in Appendix A. The optimal parameters selected for each model by the randomized grid search can be found in Appendix B.

### 3.3.6. NO$_2$ Validation Metrics

So far, we have been measuring models' performance based on manually created labels. To evaluate the uncertainty hidden in human labeling, the reference value is required. Due to the fact that there are no on-site emission measurements available at the scale of this analysis, it is, therefore, necessary to use a ship emission proxy to represent the reference value. Here, we propose to use a theoretically derived NO$_x$ emission proxy $E_s$ defined as follows:

$$E_s = L_s^2 \cdot U_s^3, \tag{6}$$

where $L_s$ is the length of the ship in *m* and $U_s$ is its speed in m/s. The details of the derivation of the given measure can be found in [11], where the proxy was introduced. As is noted in [11], the advantage of $E_s$ in comparison to other ship emission proxies (e.g., [46]) is that it can be calculated based on AIS data only, while other existing emission proxies require ship information that is not in the AIS data and is not available publicly.

The ship emission proxy is calculated for each ship of the test sets (see Figure 10). We compared the obtained values of the emission proxy with the estimated on the basis of segmentation results amount of produced $NO_2$. We estimated the amount of produced $NO_2$ by summing up $NO_2$ concentration within the pixels classified as a "plume" by each of the studied models. For the comparison between the emission proxy and the estimated amount of $NO_2$, Pearson linear correlation was used.

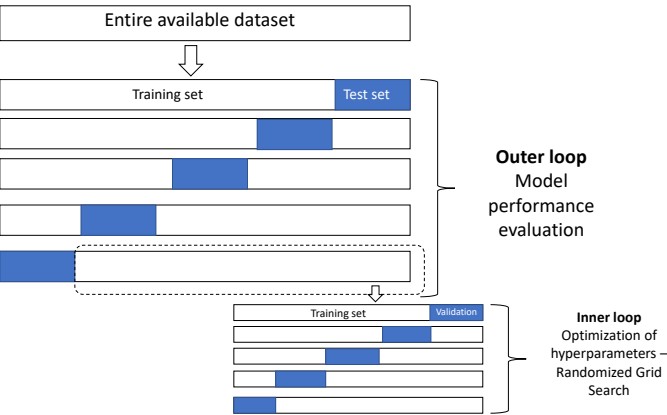

**Figure 10.** Nested cross-validation—illustration scheme.

## 4. Results

In this section, we present the results of our study. We begin with the presentation of the results of the plume segmentation model in Section 4.1. Appropriate segmentation quality is necessary for a correct estimation of $NO_2$ produced by ships. In Section 4.2, we validate the concept presented in this paper. In the Subsection, we compare the obtained on the basis of segmentation model results of ship $NO_2$ estimation with the theoretical ship emission proxy.

### 4.1. Plume Segmentation

In Table 3, we report the results of the pixel classification based on a five-fold cross-validation for all models and benchmarks studied. Figures 11 and 12 provide the precision–recall and the ROC curves respectively. Both figures were obtained by averaging the scores over all cross-validation test sets. In Figure 13, we visualize the model coefficients for the linear models studied, as well as the impurity-based feature importance coefficients for the tree-based models (Random Forest and XGBoost). The obtained results can be summarized as follows:

(i) From Table 3, Figure 11, as well as Figure 12, we can conclude that nonlinear classifiers clearly outperform both linear classifiers and threshold-based univariate benchmarks. Both used measures: AP score and ROC-AUC resulted in a similar rank of the studied classifiers. With XGBoost, Random Forest, or RBF SVM models, a very high level of precision can be achieved. For the task of ship plume segmentation, our biggest interest lies in the correct segmentation of the most representative pixels of the ship plume. Thus, the obtained level of recall we consider as reasonably satisfactory. From Table 3, we can also see that the level of the standard deviation of AP scores for multivariate nonlinear models is significantly lower than for linear or univariate models. This suggests that the results obtained with the nonlinear classifiers are more robust.

(ii) From Figure 13, we can see that Linear SVM, Logistic Regression, Random Forest, and XGBoost multivariate models utilize the spatial information provided by sub-sectors and levels. The complexity of the RBF SVM model does not allow the direct calculation of the importance of the utilized features. Even though due to the different nature of the models, the coefficients' values depicted in Figure 13 cannot be compared directly, the relative differences between the models' features go along with our intuition on where the plume produced by an analyzed ship should be located within a *normalized sector*. For

instance, high negative coefficients for the linear models that correspond to the features *Level 4* and *Level 5* suggest that even if a high-value pixel does occur in those regions of the *normalized sector*, it was most probably produced by a source other than the analyzed ship. On the other hand, the high positive coefficients corresponding to a feature *Sub-sector 2* tell us that if a high-value pixel occurs in the middle of the sector, it is most probably a part of the plume produced by the studied ship.

**Table 3.** Results on the test set with 5-fold cross-validation. Bold font indicates the best-obtained result. Under the dashed line: results obtained from univariate threshold-based methods that, in this study, we considered as benchmarks.

| Model | AP | ROC-AUC |
|---|---|---|
| Linear SVM | $0.609 \pm 0.063$ | $0.935 \pm 0.009$ |
| Logistic | $0.610 \pm 0.064$ | $0.936 \pm 0.010$ |
| RBF SVM | $0.742 \pm 0.031$ | $0.951 \pm 0.008$ |
| Random Forest | $0.743 \pm 0.030$ | $0.952 \pm 0.008$ |
| XGBoost | $\mathbf{0.745 \pm 0.030}$ | $\mathbf{0.953 \pm 0.007}$ |
| $NO_2$ threshold | $0.375 \pm 0.062$ | $0.823 \pm 0.017$ |
| Moran's $I$ threshold | $0.493 \pm 0.063$ | $0.912 \pm 0.011$ |
| Moran's $I$ on high $NO_2$ | $0.607 \pm 0.056$ | $0.922 \pm 0.010$ |

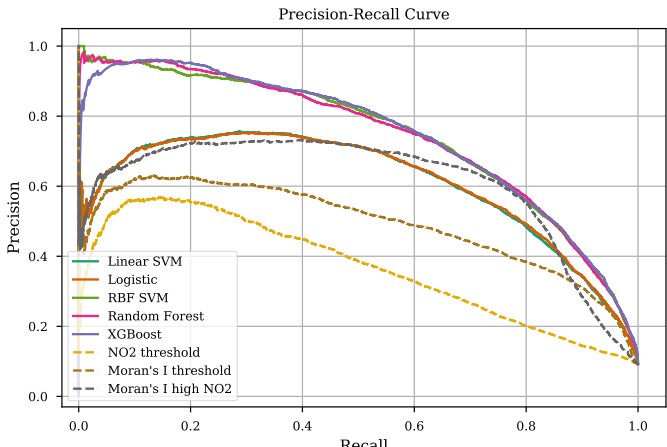

**Figure 11.** Precision–recall curve based on 5-fold cross-validation. Dashed lines indicate the results obtained from univariate threshold-based methods that, in this study, we considered as benchmarks.

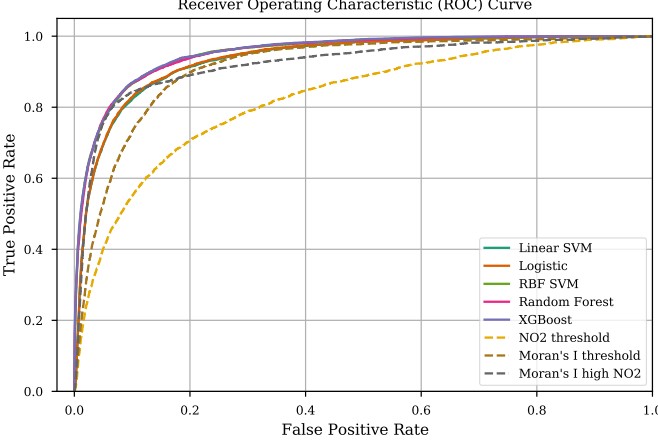

**Figure 12.** Receiver Operating Characteristic (ROC) curve based on five-fold cross-validation. Dashed lines indicate the results obtained from univariate threshold-based methods that, in this study, we considered as benchmarks.

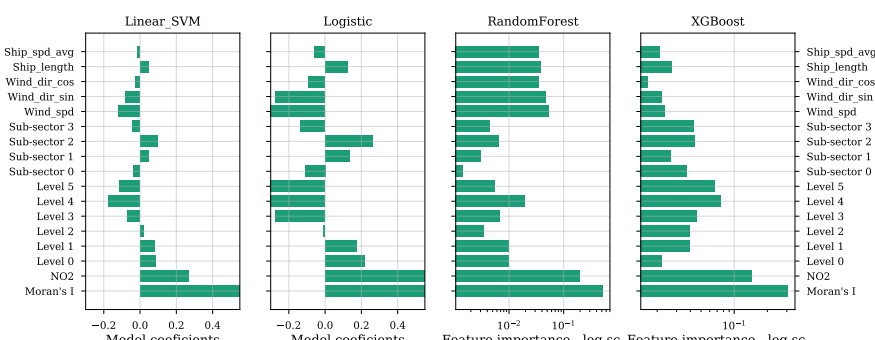

**Figure 13.** Coefficients of the features in the decision function of the linear models and the impurity-based feature importance values for tree-based models.

### 4.2. Validation with Emission Proxy

Figure 14 provides the correlation plots of $NO_2$ values estimated for a given ship on a given day based on the segmentation results of a given model and the theoretically derived $NO_x$ ship emission proxy $E_s$. Table 4 gives information on the achieved level of Pearson correlation and the number of plumes that were segmented by a certain model. Here, our baseline result is the level of Pearson correlation and the number of plumes that were identified by manual labeling. We can see that the majority of the models detected more plumes than the labeler. However, in all cases apart from XGBoost, the higher number of segmented plumes caused the decrement of the correlation score. The XGBoost model, on the other hand, was able to detect more plumes than the manual labeler, while achieving the highest correlation score. Such a result allowed us to form a hypothesis that the developed machine-learning-based methodology is able to segment plumes better than a human labeler. An example of a case where the XGBoost classifier identifies a plume better than the human labeler can be found in Figure 15. More experiments are, however, required in order to make final conclusions.

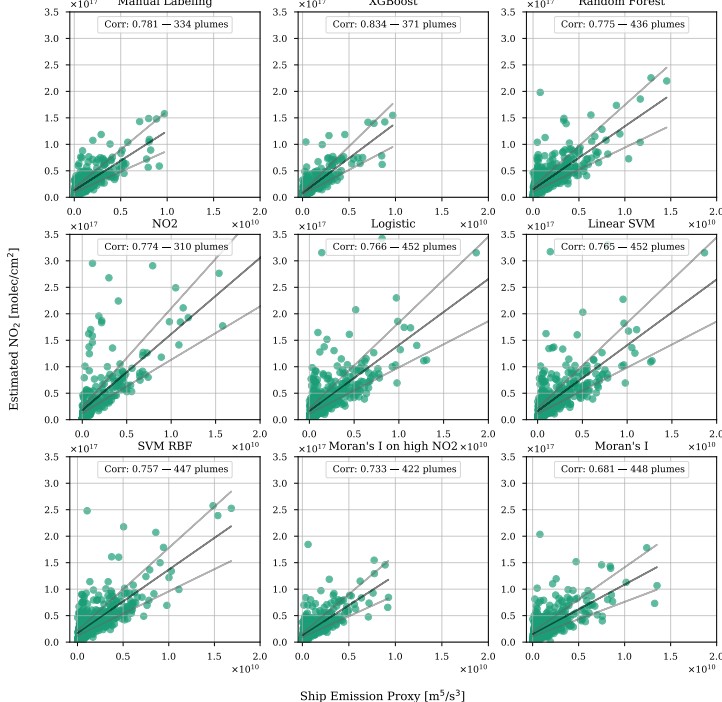

**Figure 14.** Pearson correlations between estimated (based on classification results) values of $NO_2$ emitted by each ship on a given day and a theoretical ship emission proxy. Black lines indicate a fitted linear trend. Grey lines show 30% deviations from the fitted linear trend.

The highest contrast between the scores of the performance metrics and the correlation with the emission proxy can be noted for the NO$_2$ threshold benchmark model. This is due to the fact that the ship plumes composed out of one pixel in our dataset were not labeled as plumes. The substantially high correlation with the emission proxy suggests that the single-pixel plumes were, nevertheless, identified by the method correctly. An illustration of such an example is provided in Figure 16.

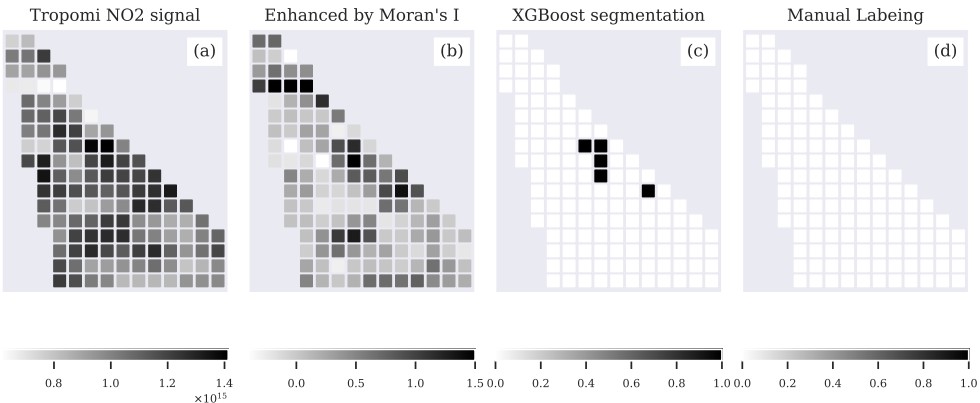

**Figure 15.** XGBoost classifier allows for the segmentation of plumes that were not recognized by the labeler. (**a**) TROPOMI NO$_2$ tropospheric vertical column density. Units: mol/m$^2$. The variable was a part of the input to machine learning models. The ship plume is difficult to distinguish by the human eye. (**b**) TROPOMI NO$_2$ image enhanced by Moran's *I*. The variable was a part of the input to machine learning models. After enhancement, the ship plume can be recognized better. At the top of the *ship sector* can be found an example when a cluster of low-value NO$_2$ was enhanced incorrectly. (**c**) Results of the segmentation of the XGBoost model. Black pixels indicate pixels classified by the model as a "plume". (**d**) Human labels. The absence of black pixels means that there were no pixels within the area labeled as a plume. For all presented images, the size of the pixel is equal to 4.2 × 5 km$^2$. Measurement date: 24 June 2019. Ship type: tanker. Ship length: 230 m. Average speed within the studied time scope: 14.27 kt.

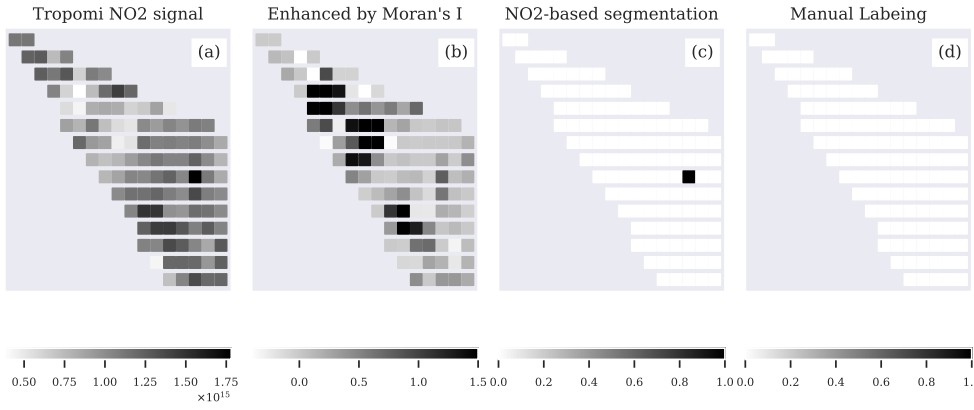

**Figure 16.** NO$_2$-based thresholding allows for distinguishing plumes cumulated within one pixel of the TROPOMI image. (**a**) TROPOMI NO$_2$ tropospheric vertical column density. Units: mol/m$^2$. (**b**) TROPOMI NO$_2$ image enhanced by Moran's *I*. At the top left of the *ship sector* can be found an example when a cluster of low-value NO$_2$ was enhanced incorrectly. (**c**) Results of the segmentation of the *NO$_2$ threshold* method. A black pixel is a pixel that was identified by a model as a plume. (**d**) Human labels. The absence of black pixels means that there were no pixels within the area labeled as a plume. For all presented images, the size of the pixel is equal to 4.2 × 5 km$^2$. Measurement date: 9 June 2019. Ship type: tanker. Ship length: 285 m. Average speed within the studied time scope: 15.4 kt.

**Table 4.** Results on the comparison between the estimated amount of $NO_2$ and theoretically derived $NO_x$ ship emission proxy. Sorted in accordance with the achieved level of the Pearson correlation. Italic font indicates baseline results.

| Segmentation Method | Pearson Correlation | Number of Detected Plumes |
|---|---|---|
| XGBoost | 0.834 | 371 |
| *Manual Labeling* | *0.781* | *334* |
| Random Forest | 0.775 | 436 |
| $NO_2$ | 0.774 | 334 |
| Logistic | 0.766 | 452 |
| Linear SVM | 0.765 | 452 |
| RBF SVM | 0.757 | 447 |
| Moran's $I$ on high $NO_2$ | 0.733 | 422 |
| Moran's $I$ | 0.681 | 448 |

## 5. Conclusions

In this study, we presented a new supervised-learning-based method for the automatic evaluation of emission plumes produced by individual ships using satellite data. The experiments were performed using $NO_2$ measurements from the TROPOMI/S5P satellite. We started with the enhancement of the satellite data in order to increase the contrast between the ship plume and the background. The applied image pre-processing technique enhances the intensity of high-value pixels located in a cluster (plume) and suppresses random concentration peaks in the background. We then automatically assigned a *ship sector* to each analyzed ship, which excludes from the analysis parts of the image where the plume of the studied ship cannot be located based on wind conditions and the speed/direction of the ship.

As a next step, we presented a feature construction method consisting of the normalization of the *ship sector* and its division into smaller sub-regions. Each sub-region has a different probability to contain a plume produced by the ship of interest. This way, we differentiated the plume produced by the ship of interest from all the other plumes potentially located within the *ship sector*. The set of newly created spatial *ship-sector*-based features allowed us to perform ship plume segmentation using multivariate machine learning models. The application of the multivariate models gives the possibility to support the ship plume segmentation process with a set of additional one-dimensional features such as ship characteristics and speed.

We integrated several data sources into a multivariate dataset. We manually labeled the data, so that the problem of individual ship plume segmentation can be addressed with supervised learning.

We trained a set of robust linear and nonlinear multivariate classifiers and compared their performance with the segmentation results of thresholding-based univariate benchmarks. All studied nonlinear classifiers showed superior results in comparison to both linear models and univariate benchmarks. With the XGBoost model, we were able to achieve more than a 20% increase in the segmentation average precision in comparison to the best benchmark univariate model.

We validated the proposed methodology using an independent measure, i.e., a theoretically derived $NO_x$ ship emission proxy that we used as a reference value. For the comparison, we estimated the amount of $NO_2$ produced by each of the analyzed ships and calculated the Pearson correlation of the obtained results with the ship emission proxy. We compared the obtained correlations and the number of plumes segmented by each of the studied models with the results obtained from manual segmentation. We showed that, with the XGBoost model, we are able to segment more plumes while achieving a 6.8% higher correlation with the emission proxy than when the plumes were segmented manually. That might suggest that the proposed method is able to find plumes that are hardly or not detectable by the human eye.

## 6. Discussion

The presented approach opens new perspectives for the application of remote sensing in the domain of ship emission monitoring. However, there are several points on the generalization of the results, the methodology, and the TROPOMI detection limit we would like to address here.

Firstly, we would like to discuss the possibility of the application of the proposed methodology to other regions. In this study, we presented a general approach that allows for the application of machine learning models for more efficient, automated segmentation of plumes from individual ships using TROPOMI data. All steps of feature preparation can be performed on the data from any region of the globe. Nevertheless, the machine learning models will have to be retrained on the region-specific datasets.

Secondly, not all regions will be equally suitable for the performance of ship emission monitoring with remote sensing. In particular, at the moment, there is no scientific evidence that, under the thick layer of land-based emission outflow, it will still be possible to differentiate plumes produced by ships. Therefore, areas that lie in close proximity to big cities, ports, or industrial objects are currently challenging to analyze.

The next point is the validation approaches used in this study. For the training of the machine learning model, we used human labels. Human labeling is the basis of all machine learning methods, and this study pioneers ship plume segmentation with more efficient supervised learning based on human labeling. However, the dispersion and chemical transformation of a ship plume, as well as its non-rigid structure mean that there are always some parts of this plume that are at or beyond the visible detection limit of the combination of the TROPOMI instrument and the retrieval algorithm. This can cause errors in labeling, as is demonstrated in Figure 15. Such mistakes if present in reasonable amounts should not affect the performance of the model, but, if the number of labeling errors is too high, the machine learning model will not be able to learn properly and, thus, the resulting performance will be very poor. The fact that nonlinear models were able to easily outperform thresholding-based benchmarks suggests that the models were able to use the provided labels for training, and thus, the labeling error rate was low. Nevertheless, an independent measure of the method evaluation is needed. Since the interest of our study centers on seagoing ships, the in situ measurements cannot be considered as a potential way of method validation. The option of on-board measurement of fuel samples cannot be performed at the scale of the study. Therefore, a theoretical measure of ship emission potential, which is the ship emission proxy, turns out to be the only available option of a reference value for the results of this study.

The usage of the ship emission proxy, however, has its limitations, namely the used ship emission proxy does not take into account many factors that influence the expected level of emission for a given ship. Nonetheless, the used proxy allowed us to rank the emission potential of the analyzed ships properly.

Following this, we would like to discuss the fact that, even though only fast ships were taken into consideration in this study, the number of ships for which the plume was possible to distinguish is higher than the number of ships for which the plume was invisible for the labeler. This study focused on observing emission sources at the edge of the detection limits of the TROPOMI instrument. It is, therefore, likely that, under certain circumstances, ship plumes remain undetected. We can only in part explain under what circumstances plumes are not visible. With the data presented in Figure 17 and Table 5, we show that, as expected, smaller and slower ships are more often not detected. Similarly, for high wind speeds, the detection is more challenging due to the high dilution of the ships' emissions and, therefore, low concentrations (the evidence can also be found in Figure 17 and Table 5). Regarding the lower detectability at lower wind speeds that can also be observed in Figure 17, we find some accordance with the findings from [47], where it is described how the wind speed impacts the reflectivity of the sea surface due to the shape of the waves, which in turn influences the sensors' sensitivity. However, this topic needs further study in the satellite retrieval community.

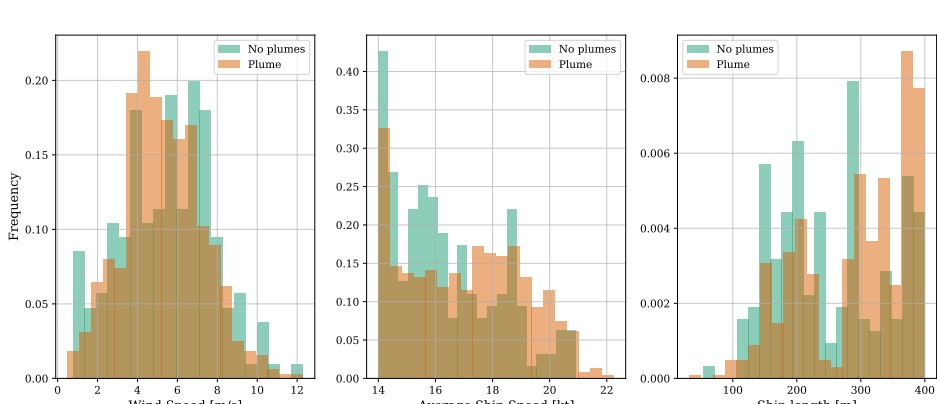

**Figure 17.** Distribution of the dataset features for the images, where there were no visible ship plumes distinguished, and for the images, where there was a visually distinguishable ship plume.

**Table 5.** Average and standard deviation for the dataset features for the images, where there were no visible ship plumes distinguished, and for the images, where there was a visually distinguishable ship plume.

| Variable Name | No Plume Image | Image with a Plume |
|---|---|---|
| Wind speed (m/s) | $5.47 \pm 2.31$ | $5.27 \pm 2.00$ |
| Ship speed (kt) | $16.83 \pm 2.01$ | $17.41 \pm 2.04$ |
| Ship length (m) | $279.92 \pm 86.64$ | $303.99 \pm 82.79$ |

To sum up, the method presented in this study is a big step towards automated and global ship emission monitoring with remote sensing and should not be devalued by the above-mentioned limitations. Firstly, one can train a machine learning model per region as commonly done in remote sensing. In addition, the region can serve as a feature of the model itself to make it invariant to geographic locations. Moreover, adding of such variables such as month, solar radiation, or temperature will make the model invariant to the seasonal changes that might be more severe at northern latitudes. Secondly, main ship routes go through both more and less suitable regions for the satellite observations. Thus, a selection of the more convenient regions will still allow us to use our approach for efficient monitoring of the emission levels produced by ships that follow those routes. Moreover, the obtained good results both in terms of segmentation quality and comparison with the emission proxy suggest that the labeling was of substantial quality. The proposed methodology also opens new research directions. For instance, human labeling can be replaced with chemical plume dispersion models, which will further improve the labeling quality and make the proposed methodology even more effective. Finally, the problem of the visibility of ship plumes that have been unrevealed with the presented study, once solved, will give us a great overview of the capabilities of TROPOMI sensors.

**Author Contributions:** Conceptualization, S.K., J.v.V and C.J.V.; methodology, S.K., J.v.V and C.J.V.; software, S.K.; validation, S.K.; formal analysis, S.K.; investigation, S.K.; resources, S.K., J.v.V and F.J.V.; data curation, S.K. and J.v.V.; writing—original draft preparation, S.K.; writing—review and editing, C.J.V., J.v.V., J.J.M. and F.J.V.; visualization, S.K.; supervision, C.J.V., J.v.V., F.J.V. and J.J.M.; project administration, F.J.V.; funding acquisition, J.v.V. All authors have read and agreed to the published version of the manuscript.

**Funding:** This work is funded by the Netherlands Human Environment and Transport Inspectorate, the Dutch Ministry of Infrastructure and Water Management, and the SCIPPER project, which receives funding from the European Union's Horizon 2020 research and innovation program under Grant Agreement No. 814893.

**Data Availability Statement:** The TROPOMI/S5P data are freely available via https://s5phub.copernicus.eu/ 28 November 2022. Starting from the product version upgrade from 1.2.2 to 1.3.0, which took place on 27 March 2019, the ECMWF operational model analyses 10 m wind data for coinciding time are available as a support product in the TROPOMI/S5P data file. For the scope of this study, the AIS data, as well as information about the dimensions of the ships were provided to us by the ILT, which is the Dutch national designated authority for shipping inspections, is participating in this research, and has access to commercial databases of AIS data and official ship registries.

**Acknowledgments:** We would like to express our gratitude to Robert Hoek (ILT) for his valuable comments and suggestions on improvements of the presentation of this article.

**Conflicts of Interest:** The authors declare no conflict of interest.

## Abbreviations

The following abbreviations are used in this manuscript:

| | |
|---|---|
| S5P | Copernicus Sentinel 5 Precursor satellite |
| $NO_2$ | Nitrogen dioxide |
| ECMWF | European Center for Medium-range Weather Forecast |
| AIS | Automatic Identification System |
| ILT | Human Environment and Transport Inspectorate of the Netherlands |
| ROI | Region Of Interest |
| SVM | Support Vector Machine |
| RBF SVM | Support Vector Machine with Radial Basis Kernel |
| XGBoost | Extreme Gradient Boosting |
| AP | Average Precision |

## Appendix A. Hyperparameters Settings

Below, the reader can find the hyperparameters' search space that was used for the optimization of the multivariate model's performance, along with the hyperparameters that were always used for the model training.

- **Linear SVM**(*random_state* = 0)
    - C: $(2 \times 10^{-2}, 2 \times 10^{-1}, 2 \times 10^{0}, 2 \times 10^{1}, 2 \times 10^{2})$
- **Logistic**(*solver* = 'saga', *l1_ratio* = 0.5, *random_state* = 0)
    - *penalty*: ('l1', 'l2', 'elasticnet', 'none')
    - C: (0.0001, 0.001, 0.1, 1)
    - *max_iter*: (100, 120, 150)
- **RBF SVM**(*kernel* = 'rbf', *gamma* = 'scale', *random_state* = 0)
    - C: $(2 \times 10^{-2}, 0.5 \times 10^{-1}, 1 \times 10^{-1}, 1.5 \times 10^{-1}, 2 \times 10^{-1}, 2.5 \times 10^{-1}, 2 \times 10^{0})$
- **Random Forest**(*n_estimators* = 500, *oob_score* = True, *random_state* = 0)
    - *min_samples_leaf*: [2; 36]
    - *max_features*: ('sqrt', 0.4, 0.5)
    - *criterion*: ('gini', 'entropy')
- **XGBoost**(*objective* = 'binary:logistic', *eval_metric* = 'aucpr', *n_estimators* = 500, *booster* = 'gbtree', *random_state* = 0)
    - *gamma*: [0.05; 0.5]
    - *max_depth*: (2, 3, 5, 6)
    - *min_child_weight*: (2, 4, 6, 8, 10, 12)
    - *subsample*: [0.6; 1.0]
    - *colsample_bytree*: [0.6; 1.0]
    - *colsample_bylevel*: [0.6; 1.0]
    - *learning_rate*: (0.001, 0.01, 0.1, 0.2, 0.3, 0.4)
    - *reg_alpha*: $(0, 1 \times 10^{-5}, 5 \times 10^{-4}, 1 \times 10^{-3}, 1 \times 10^{-2}, 1 \times 10^{-1}, 1 \times 10^{0})$

## Appendix B. Hyperparameters Selected by the Optimization Process

Here, we provide the set of the hyperparameters that were identified as optimal through the performance of the randomized grid search procedure:

- **Linear SVM:** In every iteration of the cross-validation procedure, the optimal value of parameter C was: C = 0.02.
- **Logistic regression:** In every iteration of the cross-validation procedure, the optimal value of parameter C was: C = 0.001.
- **SVM RBF:** In every iteration of the cross-validation procedure, the optimal value of parameter C was: C = 0.1.
- **Random forest:**
  - CV0: $criterion = entropy, max\_features = 0.4, min\_samples\_leaf = 18$
  - CV1: $criterion = entropy, max\_features = sqrt, min\_samples\_leaf = 24$
  - CV2: $criterion = entropy, max\_features = 0.4, min\_samples\_leaf = 24$
  - CV3: $criterion = entropy, max\_features = sqrt, min\_samples\_leaf = 18$
  - CV4: $criterion = entropy, max\_features = 0.4, min\_samples\_leaf = 18$
- **XGBoost:**
  - CV0: $gamma = 0.2, max\_depth = 6, min\_child\_weight = 10, subsample = 0.6, colsample\_bytree = 0.89, colsample\_bylevel = 0.79, learning\_rate = 0.01, reg\_alpha = 1e-05$
  - CV1: $gamma = 0.15, max\_depth = 5, min\_child\_weight = 2, subsample = 0.6, colsample\_bytree = 0.89, colsample\_bylevel = 0.6, learning\_rate = 0.01, reg\_alpha = 0.01$
  - CV2: $gamma = 0.2, max\_depth = 6, min\_child\_weight = 2, subsample = 0.6, colsample\_bytree = 0.89, colsample\_bylevel = 0.6, learning\_rate = 0.01, reg\_alpha = 0.0005$
  - CV3: $gamma = 0.2, max\_depth = 6, min\_child\_weight = 2, subsample = 0.6, colsample\_bytree = 0.89, colsample\_bylevel = 0.6, learning\_rate = 0.01, reg\_alpha = 0.0005$
  - CV4: $gamma = 0.15, max\_depth = 5, min\_child\_weight = 2, subsample = 0.6, colsample\_bytree = 0.89, colsample\_bylevel = 0.6, learning\_rate = 0.01, reg\_alpha = 0.01$

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
