# Peer review of "Supervised Segmentation of NO2 Plumes from Individual Ships Using TROPOMI Satellite Data"

_remotesensing, doi:10.3390/rs14225809_

Round 1

Reviewer 1 Report

Dear Authors,

You work is of outmost interest and will significantly add to the applicability of satellite observations of air quality. However, the paper does not sufficiently describe all the important points that affect your results, nor are the actual results properly given to the reader. Please go through my very detailed annotated document for comments and suggestions to improve this work and its future applications. 

I recommend reconsider with major revisions however I am not at all confident that you can address all the concerns within the short timeline typically permitted by MPDI. I urge you to request a sufficient amount of weeks to re-assess your work.

Congratulations and good luck!

Author Response

Dear Reviewer,

We would like to thank you for such a detailed review. We highly appreciate all the comments and suggestions provided. As all of the authors are from a data science background, we found very valuable suggestions that helped to make our article more accessible to the reader of the Remote Sensing Journal. We tried our best to implement as many changes as possible to improve our work. Please find attached our replies to the provided comments.

Sincerely,

The Authors

Reviewer 2 Report

This manuscript is based on supervised-learning-based method for an automated evaluation of emission plumes produced by individual ships using the TROPOMI/S5P satellite. Although, the authors are trying to provide a rational approach irrespective of the inherent monitoring limitations of Copernicus and the course resolution ECMWF wind data. Yet, this is systematic work that merit publication with the hope that this work acts as stimulus for further monitoring advancements. In reality, I am suggesting acceptance by the journal following adoption of all of the next comments.

Ln.1-2 and 19-21: This is a scientific publication and should be considered as such in a peer review of a scientific journal. If the authors would like to present this as compliance to regulation for a mickey mouse authority as is the IMO the United Nations and European Union then they should seek to publish it a political daily newspaper or the pamphlets of such institutions. Hence, I strongly advise the authors to stick to the scientific purposes of their work and eliminate all references to such ephemerae regulations. If it was the later I would have refused to read further and rejected this work from this serious journal.  It is more than sufficient to mention in the end that they have received funding from a Horizon 2020 project.

At Fig.1 at the caption the authors should mention what was assumed for the vertical profiles of the NO2 concentrations.

Ln.164: Why the 10m height was selected and it should be explained how many square km are in the cells of 0.25 geographical degrees. As for the 6 hours intervals the authors need to comment if the winds are expected to stay constant for such periods in the domains examined.

At Fig.2 the size of the pixels should be identified at the caption.

At Fig.5 the size of the pixels should be mentioned at the caption and the period of the year should be also incorporated. I would also like to draw the attention of selecting this region of North Africa and South Crete. This is an area effected by strong winds in particular to the vertical direction in most periods of the year. Hence, the meteorology in their approach needs further verification with actual measurements.

At all parts of Fig.9 to insert the +-30% lines for quantification of the success of relevant approach.

At the captions of Fig.10 and Fig.11 the size of the pixel in Km should be specified.

Ln.425-430: Are important and worth enhancing and incorporating at the conclusions (uncertainty after Ln.471) and the abstract.

Ln.481: Worth expanding further and quantifying the text “significant improvement in segmentation 481 quality in comparison to the existing methods”, especially if a scientist is called to defend a compliance monitoring into a court case system.

Author Response

Dear Reviewer,

We would like to thank you for the provided comments and suggestions. We tried our best to address in our work all the provided suggestions. In the attachment, you can find our replies to the provided comments.

Sincerely,

The Authors

Author Response

Dear Reviewer,

We would like to thank you for your detailed review. We highly appreciate all the comments and suggestions provided. We tried our best to implement as many changes as possible to improve our work. In the attachment, you can find our replies to the provided comments.

Sincerely,

The Authors

Reviewer 4 Report

This study showed a methodology for automated segmentation of NO2 plumes produced by seagoing ships using multivariate supervised learning on TROPOMI data. The work is uesful and interesting. However, the writing style does not meet the writing standards of journals. This makes it difficult for readers to understand and repeat the content of the article. For example, the discussion section is missing.

Author Response

Dear Reviewer,

Thank you for your comments. We made a significant effort to ensure that our paper is reproducible. Upon publication of the article, both the generated dataset and the code used for the experiments will become publicly available. We also added a section Discussion to our paper.

Sincerely,

the Authors

Round 2

Reviewer 1 Report

Dear authors,

I would like to congratulate you for improving your manuscript so greatly. The new Figures are excellent, well informative, and the updated methodology/conclusions/discussion sections really cover all the issues I had with the original version. I hope to see more published works from this team in the future. 

I recommend publish as is, after a minor English check is performed just in case something was missed. 

Best wishes,

Author Response

Dear Reviewer,

We would like to thank you for your positive feedback!  We are glad that we were able to improve the quality of the submitted work. In accordance with your suggestions, we did the additional proof checks and corrected some minor mistakes. No major changes were made since the previous revision round. 

Sincerely,

The Authors

Reviewer 4 Report

The author solved my problem. This job is great.

Minor comments:

The discussion section is usually put before the Conclusions section.  

Author Response

Dear Reviewer,

We would like to thank you for the positive feedback! We are glad that this time the work looks better. We agree that usually the Discussion section is positioned before the conclusions. In this work, however, we decided to let the order as it is as we believe, this way, the logical flow of the ideas we would like to pass to the reader is preserved better.

Thank you for understanding!

Sincerely,

The Authors